# Self-supervised Semantic Texture Decomposition for Ultrasound Segmentation and Analysis

**Tal Grutman**[1] [ID] · TALGRUTMAN@MAIL.TAU.AC.IL

**Tali Ilovitsh**[1,2] [ID] · ILOVITSH@TAUEX.TA.AC.IL

[1] *School of Biomedical Engineering, Tel-Aviv University, Israel*

[2] *The Sagol School of Neuroscience, Tel-Aviv University, Israel*

**Editors:** Accepted for publication at MIDL 2025

## Abstract

Ultrasound is an especially challenging modality to interpret because it requires unique domain knowledge to draw conclusions on the anatomy based on analysis of B-mode intensity. For this reason, there is great value in transforming B-mode images to a color scheme that is more closely aligned with the anatomy and its unique tissue properties like speed-of-sound and scattering coefficient to simplify ultrasound analysis. In this work, we introduce texture ultrasound semantic analysis (TUSA), a self-supervised transformer model trained to decompose B-mode ultrasound into distinct channels that are defined by the texture they represent. We train our model on 10 freely available ultrasound datasets and demonstrate superior segmentation performance and consistency compared to training on B-mode intensity on an additional 11th dataset. We conclude that by incorporating TUSA into the training pipeline, downstream models can focus on recognizing the anatomy instead of extracting features from intensity.

**Keywords:** Ultrasound, self-supervised learning, segmentation

## 1. Introduction

Ultrasound imaging relies on B-mode intensity, which represents tissue properties but lacks direct anatomical correlation, making interpretation dependent on expert knowledge. In this work, we focus on the texture patterns in the image, which are closely intertwined with the anatomical structures. Previous efforts have explored classical texture analysis techniques to enhance ultrasound segmentation (Milko et al., 2008; Muzzolini et al., 1993; Id et al., 2019; Zhan and Shen, 2006), but these approaches often fail to generalize due to the high variability in B-mode images caused by low contrast and noise (Abhisheka et al., 2023), which has prompted great interest in deep learning applications for ultrasound analysis, as deep learning algorithms have been able to generalize well in a wide variety of ultrasound applications (Prevost et al., 2018; Park and Lee, 2020; Byra et al., 2018; Schein et al., 2023; Shin et al., 2024; Burgos-Artizzu et al., 2020; Simson et al., 2024). However, deep learning algorithms are bounded by the quantity and quality of their datasets, and although many ultrasound datasets are freely available online, semantic pixel-wise labels only exist for targeted applications rather than for general-purpose comprehension of images. This challenge can be alleviated by using contrastive learning methods based on SimCLR (Chen et al., 2020) to build powerful representations of data so that downstream models can focus on learning the task at hand rather than how to analyze the images. Inspired by the ability of self-supervised networks to learn deep intuition, our goal is to build a general-purpose ultrasound model that can pre-analyze B-mode images and semantically label the macro-scale texture groups to improve the accessibility of ultrasound.

## 2. Methods

### 2.1. General Framework

We propose the following texture ultrasound semantic analysis (TUSA) pipeline. First, a sliding window U-Net transformer (SwinUNETR) (Hatamizadeh et al., 2022) processes an input B-mode ultrasound image with an output dimension corresponding to the expected number of unique textures visible in the dataset and is numerically encouraged to select a single texture channel using the differentiable Sparsemax activation (Martins and Astudillo, 2016). Finally, a depth-wise separable convolution reproduces the B-mode image from its texture components. After training, the segmentation model can be used as a standalone pre-processing mechanism for downstream tasks.

### 2.2. Training Data

Our TUSA model is trained on freely available ultrasound data from various sources, countries, hardware and a uniform sampling of the following organs: knee (Yerich et al., 2020), fetal (Lu et al., 2022; Burgos-Artizzu et al., 2020), cardiac (Leclerc et al., 2019), carotid (Momot, 2022), breast (Al-Dhabyani et al., 2020; A., 2024), thyroid (Pedraza et al., 2015), liver (Byra et al., 2018), and uterine fibroid (Yang, 2023). An additional dataset featuring kidney ultrasound (Singla et al., 2023) was used for downstream validation experiments in addition to the JNU-IFM (Lu et al., 2022) dataset that was a part of the training set.

### 2.3. Downstream Segmentation

Downstream Segmentation. After selecting a model candidate, we trained segmentation models on the JNU-IFM and kidney datasets. The JNU-IFM dataset contains semantic labels of the fetal head and symphysis pubis. The kidney dataset contains two types of semantic labels: a binary mask encapsulating the entire kidney and a multi-class mask of the cortex, medulla, and central echogenic complex. Each of the downstream datasets was split into training and test data, each comprising 80% and 20% of the data accordingly. We trained Attention U-Net (Oktay et al.) 100 times, half of the train runs operated on the B-mode image and half of them learned to segment the kidney from TUSA's texture logits.

## 3. Results and Discussion

Table 1: Dice scores for segmentation models trained on the kidney and JNU-IFM datasets

| Dataset | Standard B-mode | TUSA |
| --- | --- | --- |
| JNU-IFM (multi-class) | $0.87 \pm 0.02$ | $\mathbf{0.88 \pm 0.00}$ |
| Kidney (multi-class) | $0.54 \pm 0.03$ | $\mathbf{0.55 \pm 0.01}$ |
| Kidney (capsule) | $0.8 \pm 0.04$ | $\mathbf{0.83 \pm 0.01}$ |

The improvement in models trained on texture channels (Table 1) is in line with our expectation that simplifying B-mode intensity makes analysis of the image easier, as down-

stream models can skip straight to identifying target structures rather than extracting features from intensity patterns. This is reinforced by looking at individual channels of the validation data Figure 1, where certain features can be found just by isolating the cor-rect channel and analyzing the shapes, as in the case of the carotid artery (1c), fetal brain ventricle (1c), and breast lesions (1g).

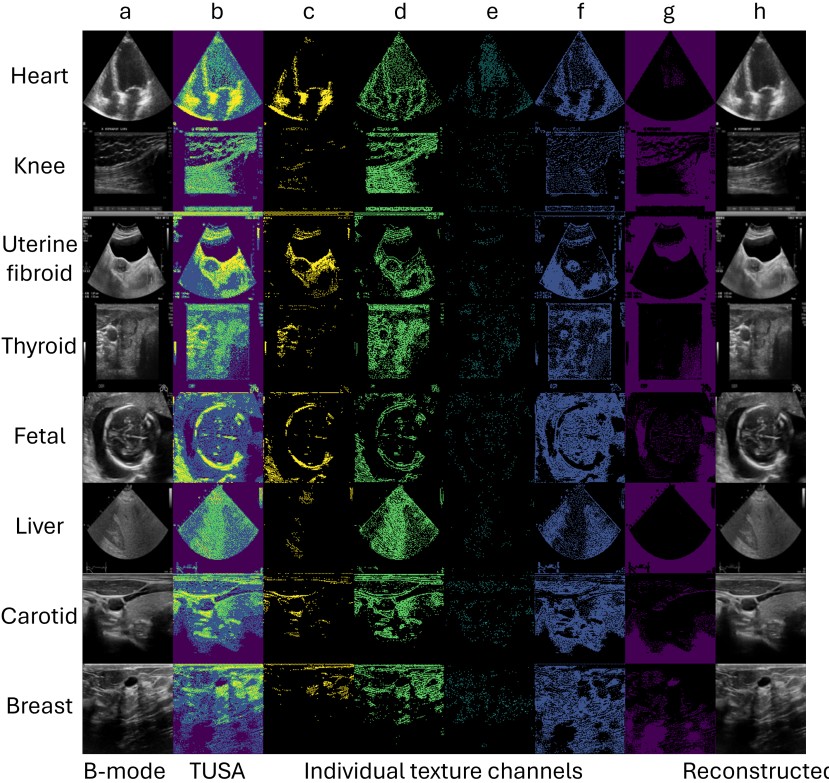

Figure 1: TUSA Validation Images– A sample image is shown for each organ in the validation set (a) alongside the corresponding TUSA decomposition (b). The individual texture channels (c-g) are presented in order of echo-genicity from left (most echogenic) to right (most anechoic). Output b-mode reconstruction (h) from TUSA feature channels.

## 4. Conclusion

We introduce TUSA, a self-supervised texture decomposition model for ultrasound segmentation and analysis. By learning a texture-based representation of ultrasound images, TUSA improves segmentation accuracy and consistency across datasets. Our approach provides a general-purpose tool for ultrasound analysis, reducing the reliance on expert annotations and enhancing model interpretability.

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
