# OpenReview forum: "Self-supervised Semantic Texture Decomposition for Ultrasound Segmentation and Analysis"
_MIDL.io/2025/Short_Papers — MIDL 2025 - Short Papers_

### Official Review · Reviewer_jJWg · 2025-04-24

**Rating:** 4
**Confidence:** 5

**Summary:**

The authors address the challenge of interpreting B-mode ultrasound images due to their reliance on intensity values, which may not directly correspond to anatomical properties. They introduce Texture Ultrasound Semantic Analysis (TUSA), a self-supervised transformer model designed to decompose B-mode ultrasound into distinct channels based on texture representations, aiming for a representation more aligned with tissue properties. Trained on 10 public ultrasound datasets, TUSA's effectiveness was evaluated by using its output channels as input for downstream segmentation tasks on an 11th dataset.

**Strengths:**

The proposed method (TUSA) for extracting texture-based channels from ultrasound via self-supervision is conceptually interesting. This approach would likely be of interest to the medical image analysis community, especially those specializing in ultrasound data.

**Weaknesses:**

The actual improvements in Dice score appear marginal in magnitude, which raises questions about the practical significance of adopting the TUSA pipeline compared to using B-mode images directly for the task of segmentation.

---

### Decision · Program_Chairs · 2025-05-01

Accept